# FINEdits : Precise Image Editing with Inferred Masks and Light Fine-tuning

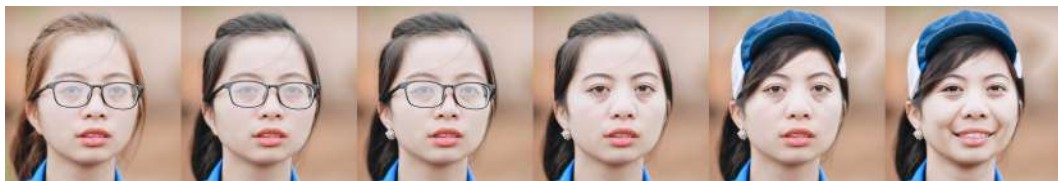

(a) A full edit sequence on a sample from our proposed EditFFHQ dataset using our editing method: dark hair, add earings, remove glasses, add a hat, smile.

Figure 1: Edit sequence demonstrations from our proposed EditFFHQ dataset

## Abstract

In image editing with diffusion models, it is a difficult challenge to achieve a balance between edit fidelity and preservation of the content which is unrelated to the editing objective. Training-free methods often suffer from imperfect inversion that degrades reconstruction quality, while training-based approaches require substantial computational resources and carefully curated datasets. We present FINEdits, a method that addresses these limitations through two key contributions: (1) we exploit cross-attention maps to define a mask which explicitly preserves non-edited regions, and (2) we use lightweight fine-tuning to improve inversion quality without semantic drift. Our masking approach leverages transformer attention mechanisms to automatically identify editing regions using a parameter-free K-means clustering method, eliminating the need for manual hyperparameter tuning. To handle the inversion quality degradation at early timesteps required for large edits, we introduce a light fine-tuning strategy that balances reconstruction fidelity with semantic preservation. Furthermore, we introduce EditFFHQ, a new benchmark dataset of 2000 face images with sequential editing instructions, enabling quantitative evaluation of identity preservation and edit quality. Extensive experiments demonstrate that FINEdits achieves superior identity preservation while maintaining competitive edit fidelity and image quality. Our method provides an effective solution for precise image editing that preserves visual consistency without requiring extensive retraining or manual parameter adjustment.

## 1 Introduction

Image synthesis from text prompts has taken incredible strides forward in recent years, with ever-increasing quality and high-level semantic control. The most recent generative models are based on diffusion (Sohl-Dickstein et al., 2015; Ho et al., 2020; Song et al., 2021b) and flow-matching methods (Liu et al., 2022; Lipman et al., 2023), using transformer-based techniques to further improve quality (Peebles & Xie, 2023; Esser et al., 2024; Labs, 2024). Diffusion-type generative models rely on a forward process which progressively adds noise to an image, and a learned reverse process which removes it. Thus, starting from pure random noise, it is possible to synthesize a random image. Text prompts are added as conditions to the generation.

These models were subsequently repurposed with great success for image *editing*. One such approach is a technique called *image inversion* (Song et al., 2021b;a). This consists in carrying out the

forward process to a certain degree of noise (also referred to as noise strength), and then proceeding with the reverse process, but then *changing* the condition, to achieve the final editing task. Intuitively, the noise strength must be chosen according to the amplitude of the edit: larger edits require greater strength. Unfortunately, this can have the undesired effect of removing structures/content which we do not wish to modify. Thus, there is a fundamental tradeoff between editing, which may require strong noise, and fidelity to the original image (also referred to as "reconstruction"), which on the contrary requires less strong noise. We refer to this tradeoff as the editability/reconstruction tradeoff, and addressing this issue is at the heart of our proposed method.

In this paper, we propose an inversion-based editing method that relies on fine-tuning and masking based on attention maps. Fine-tuning enables better fidelity (reconstruction) to the source image, and masking allows localized editing, ensuring better preservation of the essential elements of the original image. The FINEdits method is tested here on face editing. To quantitatively evaluate FINEdits on a significant dataset, we propose a labeled dataset EditFFHQ. FINEdits compares favorably to state-of-the-art methods; in particular, it demonstrates better identity preservation capability than other methods such as Kontext Batifol et al. (2025) while ensuring high editing success rates.

## 2 RELATED WORK

Research on image editing with diffusion and flow matching models has developed along two complementary directions. Training-free methods achieve editing by manipulating the generation dynamics of pre-trained models, requiring no additional training. On the other hand, training-based methods expand the model's capabilities by optimizing new parameters, either through fine-tuning or the introduction of adapter networks.

**Training-Free Methods** mostly rely on perturbing the source image via a forward noising process to project it to a more editable state, and then applying a reverse denoising process to guide the image towards a target edit. SDEdit Meng et al. (2022) was the first to demonstrate this idea to produce edits where coarse sketches were transformed into realistic samples. However, large edits require sending the source image to strong noise levels to be faithfuly applied, which significantly diminishes the fidelity to the input. A major step forward came with the introduction of deterministic solvers, such as in (Song et al., 2021b) and DDIM (Song et al., 2021a). Beyond accelerating sampling, these solvers allow for image inversion, that is, mapping the image back to its latent representation by reversing the denoising process. When the image is well aligned with the model distribution, inversion yields a likely generation trajectory that can be reused for editing, and this principle has since become the foundation of many training-free approaches. Building on this, several work refine inversion for editing. DiffEdit (Couairon et al., 2022) combines automatic mask generation with DDIM inversion, ensuring perfect reconstruction in unedited regions. Other methods exploit reference cross-attention maps to better control the layout during editing : (Parmar et al., 2023) aligns attention maps through gradient steps, while Prompt-to-Prompt (Hertz et al., 2022) directly substitutes maps from the reference prompt. Null-Text Inversion (Mokady et al., 2023) extends these techniques to real images by optimizing null embeddings during DDIM inversion. More recent approaches further improve fidelity: DDPM inversion (Huberman-Spiegelglas et al., 2024) achieves exact reconstruction by storing noise maps, while LEdits++ (Brack et al., 2024) restricts edits using masks derived from cross-attention. RFSolver (Wang et al., 2024a) increases inversion accuracy by employing higher-order ODE solvers, albeit at higher computational cost. Despite their effectiveness, training-free methods often require careful hyperparameter tuning and remain sensitive to inversion quality, which can limit their robustness and reliability in practice. FINEdits addresses the issue of inversion quality with a light single-image fine-tuning, achieving a better balance between editability and preservation of the original image, while retaining a relatively low hyperparameter count.

**Training-Based Methods** take a different approach by directly learning the parameters of the model dedicated to image editing, which generally translate to better editing performance. ICEdit (Zhang et al., 2025) leverages the inherent capability of modern text-to-image DiTs (Esser et al., 2024; Labs, 2024) to generate coherent panels and further improves the editing performance by learning LoRA Hu et al. (2022) parameters to better follow editing instructions. Another research direction focuses on instruction-based image editing. InstructPix2Pix (Brooks et al., 2023), EmuEdit (Sheynin et al., 2024), and UltraEdit (Zhao et al., 2024) curate large datasets of source images, target im-

ages, and edit instructions, allowing diffusion models to incorporate editing prompts directly during training. Flux-Kontext (Batifol et al., 2025) extends this strategy with a rectified flow transformer and concatenates text tokens from the instruction prompt with image tokens from the source image, thus allowing mutual interaction during the joint-attention operation. Finally, some methods adapt the model at edit time to better align with a specific source image. Unitune (Valevski et al., 2023) fine-tunes the noise predictor on the source image and accompanying text description, and then applies SDEdit for editing. Imagic (Kawar et al., 2023) optimizes both the prompt embedding and the model weights, enabling smooth interpolation between the source and target prompts. Training-based methods often achieve stronger semantic control and sometimes higher edit fidelity than training-free methods, but require additional training, heavier computation and rely on efficient data collection or carefully designed strategies to construct pairs of source images, edited images, and editing instructions, which further limits their scalability. Our method overcomes such burdens by fine-tuning on a single image, ensuring a well-integrated edit.

## 3 METHOD

### 3.1 BACKGROUND

**Flow matching models.** Modern text-to-image flow matching models generate samples from a data distribution $p_0$ by iteratively denoising a Gaussian noise sample $z_t$. This process is described by the *probability flow ODE*:

$$\frac{dz_t}{dt} = v(z_t, t) \tag{1}$$

where the right-hand term, called the *velocity*, can be learned via the conditional flow matching objective:

$$\mathcal{L}_{CFM} = \mathbb{E}_{t,z_0,\epsilon} \|v_\theta(z_t, t) - (\dot{\alpha}_t z_0 + \dot{\sigma}_t \epsilon)\|_2^2 \tag{2}$$

where $v_\theta$ denotes the learned velocity, $\epsilon \sim \mathcal{N}(0, I_d)$, and $z_t = \alpha_t z_0 + \sigma_t \epsilon$ is an interpolation between $\epsilon$ and $z_0$. For the linear interpolant $\alpha_t = 1 - t$ and $\sigma_t = t$, which is the most commonly used, the target velocity simplifies to $\epsilon - z_0$. The aforementioned processes are carried out in the latent space of a VAE with encoder $\mathcal{E}$ and decoder $\mathcal{D}$, such that for a given image $I$, $z_0 = \mathcal{E}(I)$ and $I = \mathcal{D}(z_0)$. Additionally, the velocity prediction $v_\theta(z_t, t)$ can be influenced by supplementary conditions such as class information or text, which we denote as $c$.

**Inversion.** A common strategy for image editing with text-conditioned flow matching models relies on inversion. In this setting, one can solve equation 1 forward in time, starting from $t = 0$, with the source latent $z_0^s$ and the source description $c^s$, up to a timestep $t_s$, referred to as the *strength*. Then, editing can be achieved by solving equation 1 again, in reverse time, starting from $z_{t_s}$ and using the target textual conditioning $c^t$ to obtain the edited latent $z_0^t$.

**Identity preservation.** For image editing on human subjects, preserving the identity is a crucial success factor. We propose to quantify this notion by leveraging embeddings produced by Arcface Deng et al. (2019), an identity classification model, which we denote as $\mathcal{A}$. Identity embeddings for an image $I$ are obtained by removing the classification MLP at the end of the model and taking the output of $\mathcal{A}(I)$. The proximity of two identity embeddings can be measured by computing their cosine similarity Thus, for a source image $I^s$ and an edited image $I^t$ we define the Identity Preservation (IP) metric as:

$$\text{IP} = \text{sim}(\mathcal{A}(I^s), \mathcal{A}(I^t)) \tag{3}$$

### 3.2 LIGHT FINE-TUNING FOR IMPROVED INVERSION

Editing large regions using inversion-based methods requires integrating equation 1 up to a large $t_s$. As previously described, this poses the problem of error accumulation when using numerical ODE solvers for the inversion. If the inversion is imperfect, crucial identity information is lost and cannot be recovered during reconstruction. We argue that part of the error induced in the inversion

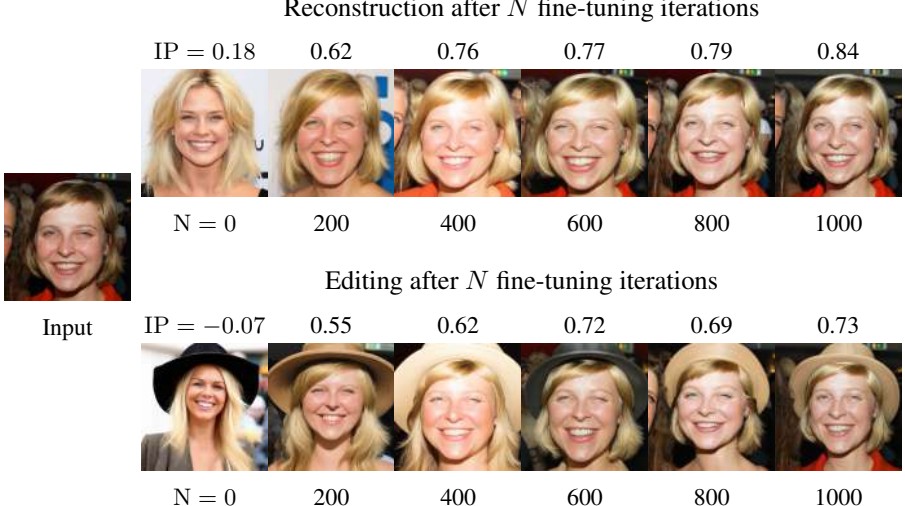

Figure 2: Impact of fine-tuning on reconstruction and editing. Inversion is applied on the input with maximum strength, using the conditioning source prompt: "A blonde woman, smiling". *Top*: Reconstruction using the same prompt.*Bottom* : Editing with the prompt"A blonde woman, smiling, wearing a hat". Fine-tuning greatly improves identity preservation, without harming editing performance.

stems from the fact that the image to edit is not a sample generated by the model and the semantic alignment between the inversion prompt and the image is suboptimal. Our method begins with single-image fine-tuning on the user-supplied image and prompt. This procedure adjusts the model such that the provided image becomes a more probable sample under the conditioning prompt. Technically, this amounts to applying the usual flow-matching objective (Eq. 2) to just one image–prompt pair.

Figure 2 demonstrates the impact of fine-tuning prior to inversion and reconstruction. The example shows both reconstruction and editing results for an image inverted with maximal strength ($t_s = 1$), as the number of fine-tuning steps increases. We observe a clear trend: additional training steps consistently enhance identity preservation, which is evident both qualitatively and through the Identity Preservation metric. Importantly, this adaptation remains lightweight and does not compromise semantic accuracy, as illustrated by the correct placement of the hat in the edited image even after $N = 1000$ steps.

### 3.3 LOCALIZED EDITING WITH INFERRED MASKS

While fine-tuning substantially improves inversion and yields a sharp increase in editing performance, it remains imperfect and introduces residual reconstruction errors. High-frequency details such as skin texture are often lost, and slight geometric variations can still appear, leading to suboptimal identity preservation. For instance, at N=1000 in the bottom row of Figure 2, the reconstructed face closely resembles the input, yet the skin tone is marginally lighter and the person on the left is missing. Comparable effects can be observed in Figure 3(a).

To address these limitations, our approach incorporates *localized editing*, which explicitly preserves regions of the image that should remain unchanged. Deciding which areas to protect and which to modify is a non-trivial challenge, particularly when relying only on textual prompts. To overcome this, we exploit internal signals from the model itself. *FINEdits* leverages the attention maps of the generative model to construct the mask $M$. Prior work has shown that the intermediate layers of transformer blocks capture rich semantic information (Tumanyan et al., 2023; Luo et al., 2023; Epstein et al., 2023; Helbling et al., 2025), which can be repurposed to guide image editing (Hertz et al., 2022; Parmar et al., 2023; Epstein et al., 2023; Brack et al., 2024).

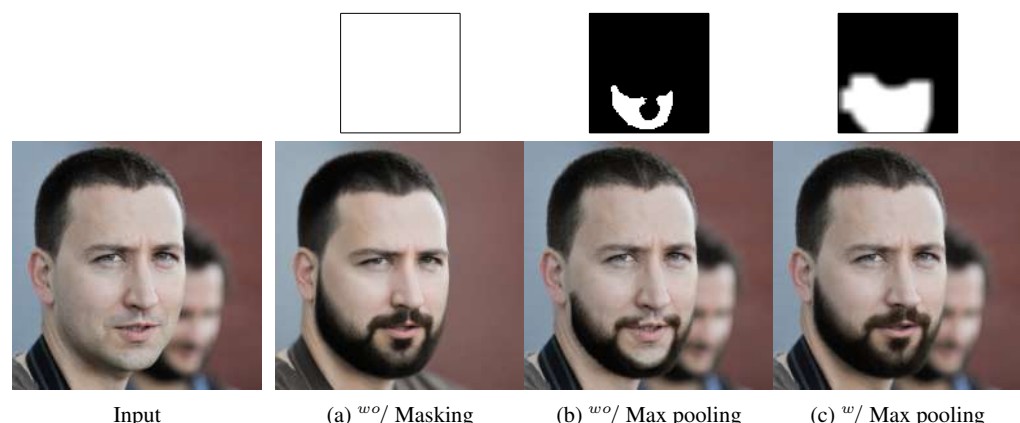

|  Input | (a) $^{wo}/$ Masking | (b) $^{wo}/$ Max pooling | (c) $^{w}/$ Max pooling |

Figure 3: Impact of masking on edit quality. *Bottom* : (a) Baseline inversion with fine-tuning 3.2 allows for a faithful edit but loses details in unedited regions, such as skin grain and the person in the background. (b) Baseline masking preserves unedited regions but slightly undershoots the area to be edited. (c) Our masking component yields faithful edits while preserving unedited regions. *Top* corresponding masks: (a) No mask (b) Mask binarized with K-means (c) Mask binarized with K-means, on top of which maxpooling and gaussian blur are applied.

The mask is derived from one of the transformer blocks cross-attention maps linking the subset of edit-related text tokens to the image tokens. Since not all transformer layers provide equally meaningful attention patterns, we empirically find that layers 8–12 of the Stable Diffusion 3 DiT contain the most useful semantic information, and we therefore select the 10th layer for mask construction. Let $c^q$ denote the subset of text tokens corresponding to the concept to be modified. To this end, we compute the attention weights $W_{c^q} = \text{softmax}(Q_i K_t[c^q])$, where $K_t[c^q]$ denotes the keys restricted to the indices of $c^q$. The resulting tensor $W_{c^q}$ has shape $N \times S_i \times |c^q|$, with N the number of attention heads, $S_i$ the sequence length of image tokens, and $|c^q|$ the number of selected concept tokens. These attention maps highlight the spatial regions most associated with the edit concept— for example, in Figure 3, $c^q$ corresponds to the tokens of the word *"beard"*. We then average the cross-attention tensor over the heads, yielding a tensor of size $S_i \times S_t$. We then restrict this tensor to the columns corresponding to the concept tokens $c^q$, and average across them. This produces a vector of length $S_i$, which encodes the average spatial attention associated with the edit concept. Reshaping this vector into a $\sqrt{S_i} \times \sqrt{S_i}$ grid yields a low-resolution attention map that highlights the regions of the image most related to the target concept. To binarize this map, we apply $k$-means clustering with two centroids, reflecting the binary nature of the editing mask. An illustration of the mask computation process is shown in Figure 4.

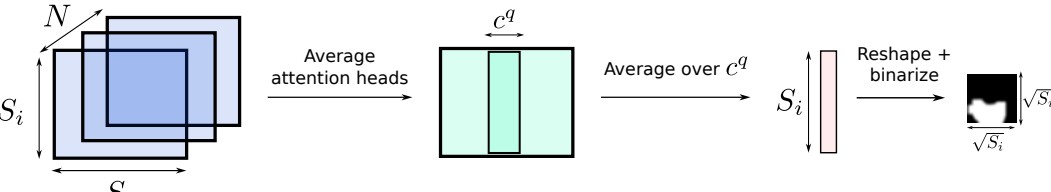

Figure 4: Mask computation

Recomputing the mask at every generation step proves ineffective, as the signal tends to vanish toward the end of the backward process. Instead, we compute it once at a specific step $t_M$, chosen such that $t_M < t_s$, ensuring that the object to be edited is not fully visible at the time of mask estimation. For concept removal tasks, this condition is naturally satisfied since the mask can be obtained during inversion. For concept addition, however, we proceed differently: the backward process is first run from $t_s$ down to $t_M$ without masking, the mask is then computed at $t_M$, and finally the process is backtracked to $t_s$ before resuming the actual editing. Formally, the final binary

mask is denoted $M \in \{0,1\}^{\sqrt{S_i} \times \sqrt{S_i}}$, where $M_{u,v} = 1$ indicates that the spatial position $(u, v)$ is associated with the concept to be edited, and $M_{u,v} = 0$ otherwise.

Couairon et al. (2022) noticed that it is beneficial to have a mask that overshoots the area to be edited. We provide such property to FINEdits by applying a max pooling operation to make it bigger. The pasting operation can sometimes leave artifacts in the final image, where the limit of the mask can be spotted. We counteract this by smoothing the mask, using a Gaussian blur operation, to make the edited and preserved parts merge more smoothly.

To summarize, let us consider the following reference inversion trajectory under $v_\theta$ and $c^s$ : $\{z_0^{ref}, z_{t_1}^{ref}, \ldots, z_{t_s}^{ref}\}$. Assuming we have a binary mask M that separates areas that should be edited from areas that must remain untouched, we propose to follow Couairon et al. (2022) and paste reference parts on the current ODE solver iterate $z'_{t-1}$: $z_{t-1} = M * z'_{t-1} - (1 - M) * z_{t-1}^{ref}$. A pseudocode for the FINEdits method can be seen in Algorithm 1.

---

**Algorithm 1:** FINEdits algorithm

---

**Input:** $v_\theta, z_0^s, c^s, c^t, t_s, \{\sigma_t\}_t$
**Output:** $z_0^t$
$v_\theta = \text{FINETUNE}(v_\theta, z_0^s, c^s)$
$z_t = z_0^s$
$z_0^{ref} = z_0^s$
**for** $t = 0, \ldots, t_s$ **do**
    $z_{t+1} = (\sigma_{t+1} - \sigma_t)v_\theta(z_t, t, c^s)$
    $z_{t+1}^{ref} = z_{t+1}$
**end**
$M = \text{GETMASK}(\{z_t^{ref}\}_t, c^s, c^t)$
**for** $t = t_s, \ldots, t_1$ **do**
    $z'_{t-1} = (\sigma_{t-1} - \sigma_t)v_\theta(z_t, t, c^t)$
    $z_{t-1} = M * z'_{t-1} + (1 - M) * z_{t-1}^{ref}$
**end**
$z_0^t = z_t$
**return** $z_0^t$

---

## 4 EXPERIMENTS

In this section, we present a comprehensive evaluation of our proposed framework. We begin by introducing EditFFHQ, a benchmark specifically designed to measure both fidelity to the original image and editability across multiple facial attributes. We then apply our method on SD3 and provide a systematic comparison against recent state-of-the-art editing methods, including UltraEdit, SDEdit, ICEdit, Kontext-dev, and RFSolver, considering both single-shot edits and sequences of edits. For methods that contain hyper-parameters, including ours, we select them using a held-out subset of EditFFHQ. Finally, we report an ablation study to quantify the contribution of each component of our approach. Additional hyperparameters and implementation details are provided in Section A.1.

**EditFFHQ.** Most image-editing evaluations focus mainly on how well the edited image aligns with the prompt and on overall image quality, typically measured with metrics such as CLIP score or FID Heusel et al. (2017). While useful, these measures ignore fidelity to the original image, making it difficult to assess whether edits preserve identity and non-edited content. To address this gap, we introduce EditFFHQ, a benchmark of 2,000 images sampled from FFHQ and annotated for editability across seven attributes: beard, hair, earrings, hair color, hat, glasses, and smile. Attribute labels are obtained automatically using `qwen-vl-27B` Wang et al. (2024b) vision–language model. We assign attributes by gender when relevant (e.g., beard and hair for men; hair color and earrings for women), and apply the remaining attributes to all. To ensure validity, we exclude children, as many edits do not apply to them. To evaluate editing performance, we report five metrics. Identity preservation is measured as described in Section 3.2. Edit success is assessed automatically

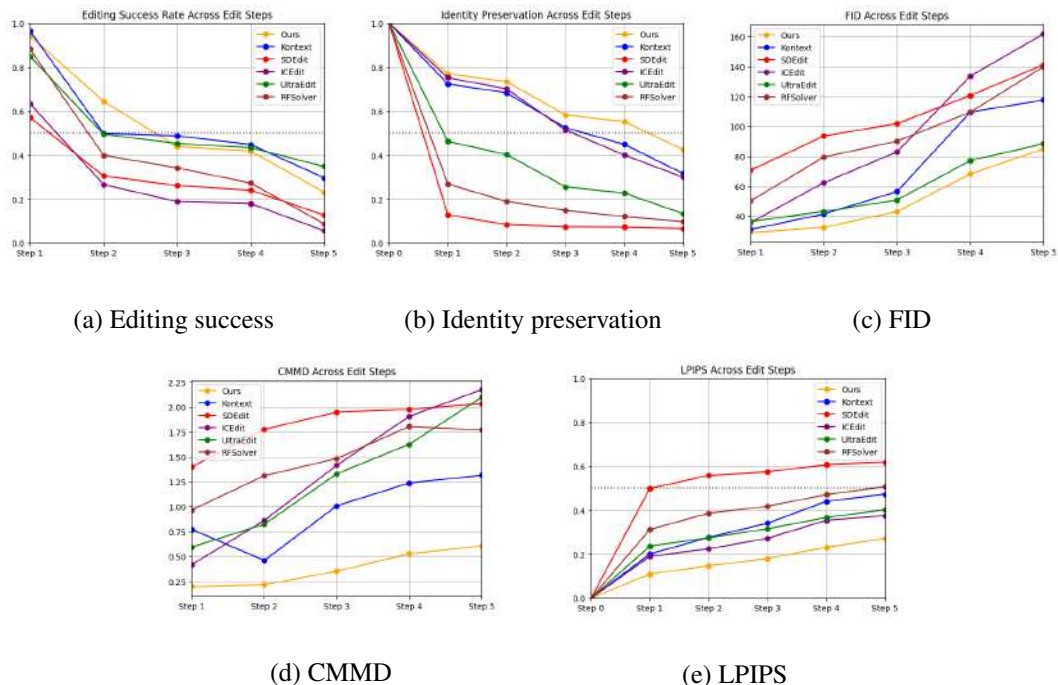

(a) Editing success      (b) Identity preservation      (c) FID

(d) CMMD      (e) LPIPS

Figure 5: Quantitative results on consecutive edits on EditFFHQ.

using the VLM as a judge. Finally, we evaluate the quality of successfully edited images with FID and CMMD Jayasumana et al. (2024), the latter having been shown to correlate more strongly with human preference.

**Single-shot editing.** Table 1 reports results on single-shot edits. To ensure a fair evaluation, we compute IP, FID, CMMD, and LPIPS only over images where the edit was successfully applied. This is important, as methods that fail to modify the target attribute often leave the image unchanged, which would trivially yield high identity preservation and low distortion. By filtering to include valid edits, both success rate and identity preservation reflect meaningful editing rather than artifacts of failure. Under this setting, our method outperforms all competing approaches by a large margin in terms of identity preservation, FID, CMMD, and LPIPS, while achieving a success rate nearly on par with Kontext-dev, a method explicitly trained for editing. These results highlight the efficiency of our framework, as it consistently surpasses larger Flux-based models despite being built on the more lightweight SD3 architecture.

| Method | IP ↑ | FID ↓ | CMMD ↓ | LPIPS ↓ | Success rate |
|---|---|---|---|---|---|
| UltraEdit | 0.46 | 36.47 | 0.60 | 0.24 | 0.85 |
| SDEdit | 0.13 | 70.73 | 1.40 | 0.50 | 0.57 |
| ICEdit | 0.75 | 35.81 | 0.42 | 0.19 | 0.63 |
| Kontext-dev | 0.72 | 31.23 | 0.77 | 0.20 | **0.96** |
| RFSolver | 0.27 | 50.22 | 0.97 | 0.31 | 0.88 |
| FINEedits (ours) | **0.77** | **29.24** | **0.20** | **0.11** | 0.94 |

Table 1: Quantitative comparison of different methods on a single edit on EditFFHQ.

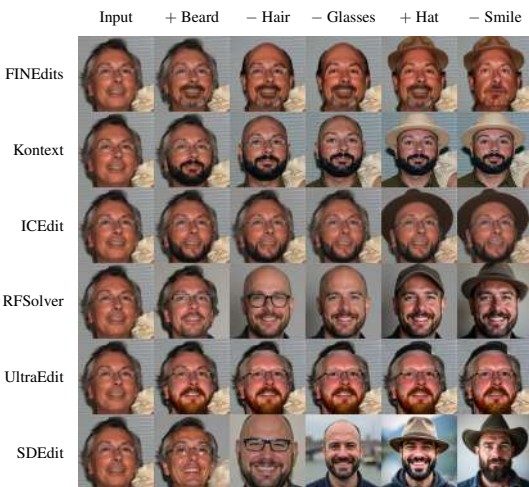

Figure 6: Qualitative comparison of different editing methods

**Sequential editing.** We further evaluate performance on sequences of five consecutive edits, always applying attribute modifications in the same fixed order for each identity. To ensure consistency, if an edit fails the sequence is terminated and we move on to the next identity. Results are shown in Figure 5. We observe trends consistent with the single-shot setting: our method achieves the best scores across all metrics, while maintaining a success rate nearly on par with the strongest competitors. Figures 6 provide a qualitative comparison. Our approach better preserves the appearance of the original identity throughout the sequence, whereas Kontext, although faithful in applying edits, tends to generate unrealistic textures as edits accumulate—an effect also reflected in its higher FID values.

**Ablation study.** To assess the contribution of each component, we perform an ablation study by removing fine-tuning and masking individually. Results are reported in Table 2. Without fine-tuning, identity preservation drops sharply (0.52 vs. 0.77), and CMMD increases, indicating weaker semantic alignment. Removing masking leads to a strong degradation in FID and LPIPS, showing that uncontrolled edits harm both realism and perceptual quality. When combined, fine-tuning and masking yield consistent improvements across all metrics, confirming that both components are essential to the performance of our method. The success rate decreases slightly when masking is applied, which is a natural trade-off: by constraining the edits to the relevant regions, the model has less freedom to satisfy the prompt but achieves far better fidelity and perceptual quality.

| Method | IP ↑ | FID ↓ | CMMD ↓ | LPIPS ↓ | Success rate |
|---|---|---|---|---|---|
| $^{wo}/$ fine-tuning | 0.52 | 32.15 | 0.88 | 0.15 | 0.99 |
| $^{wo}/$ masking | 0.62 | 34.81 | 0.36 | 0.24 | **0.96** |
| $^{w}/$ fine-tuning & masking | **0.77** | **29.24** | **0.20** | **0.11** | 0.94 |

Table 2: Ablation on the fine-tuning and masking components of our method.

## 5 LIMITATIONS

The main limitation of our method is its reliance on per-image fine-tuning. Although lightweight, this step introduces additional computational overhead which can be mitigated with parameter-efficient techniques such as LoRAs. However, the relative cost diminishes as the edit sequence grows, since subsequent edits are performed as efficiently as standard text-to-image generation. A second limitation concerns the mask computation, which depends on hyperparameters chosen based on average-case performance. While generally robust, this procedure may occasionally fail, leading to imperfect localization of the edits.

## 6 CONCLUSION

In this paper, we proposed FINEdits, an inversion-based method that combines lightweight fine-tuning with automatic masking, and introduced EditFFHQ, a benchmark for measuring both fidelity and editability with new metrics for identity preservation and edit success. Our method leverages attention maps in order to establish a mask to indicate where the editing takes place. This avoids unwanted edits in other regions, which is a common defect of editing models. FINEedits also employs a fine-tuning step to ensure high quality editing results. Our experiments on single and sequential edits show that the approach consistently outperforms recent baselines across all metrics. Our algorithm yields high quality, consistent, editing results, while maintaining strong idenitity preservation. We hope that the EditFFHQ benchmark will be a useful resource for providing rigorous evaluations of editing algorithms.

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

# A APPENDIX

## A.1 IMPLEMENTATION DETAILS

In this section we give additional information on the experimental settings for our EditFFHQ benchmark. For instruction-based methods, we constitute a pool of instructions and pick the appropriate one depending on the current attribute to be edited in the sequence.

**FINEdits.** For our method, we fine-tune the Stable Diffusion 3 DiT for 1000 steps with a learning rate of 5e-5. During early experiments we used Adam8bit optimizer so the fine-tuning could run on a RTX4090, and we kept this optimizer for the benchmark. We used a batch size of 1, with gradients accumulation of 10, logit-normal timestep sampling and linear interpolant path.

For kernel size of the max pooling operator as well as inversion strength, we find them empirically on a small subset of EditFFHQ, yielding the following edit-wise hyperparameters.

| Edit Type | Kernel Size | Inversion Strength |
|---|---|---|
| Hair color | 15 | 0.92 |
| Hat | 15 | 0.90 |
| Glasses | 21 | 0.80 |
| Hair | 11 | 0.90 |
| Beard | 25 | 0.94 |
| Earrings | 11 | 0.86 |
| Smiling | 11 | 0.72 |

Table 3: Empirically found kernel size and inversion strength with respect to the edited attribute.

**Kontext.**

For Kontext we used the following instructions : "Add hair to this man", "Make this man bald", "Add a hat to this person", "Remove the hat from this person", "Add earrings to this person", "Remove the earrings off this person", "Remove the beard", "This person now has a beard", "This person is not smiling anymore", "Make this person smile", "Take the glasses off this person's face", "This person now wears glasses", "This person now has light hair", "This person now has dark hair"

**ICEdit.**

For ICEdit, we use the following instructions : "the person now has light hair", "the person now has dark hair", "the person does not wear a hat anymore", "the person now wears a hat", "the person does not wear glasses anymore", "the person now wears glasses", "the person is now bald", "the person now has hair", "the person does not have a beard anymore", "the person now has a beard", "the person is not wearing earrings anymore", "the person is now wearing earrings", "the person is not smiling anymore", "the person is now smiling"

We do not employ the optional inference time scaling with the VLM judge.

**UltraEdit.**

For UltraEdit, we used exactly the same instructions as for Kontext.

**RFSolver.**

For RFSolver, we set features injection step to 0 for hair color, hat, hair, smiling and set it to 2 for glasses, beard and earrings. For the solver order we remain at 2 which is the default in the implementation.

**SDEdit.**

For SDEdit we use Huggingface's implementation with `flux-dev.1`. Regarding noising strengths, we found the following set of hyperparameters :

| Edit Type | Noising Strength |
|---|---|
| Hair color | 0.92 |
| Hat | 0.90 |
| Glasses (add) | 0.80 |
| Hair | 0.90 |
| Beard (add) | 0.94 |
| Earrings (add) | 0.86 |
| Earrings (remove) | 0.86 |
| Smiling (add) | 0.72 |

Table 4: Noising strength parameters for different edit operations.

