# OpenReview forum: "FINEdits : Precise Image Editing with Inferred Masks and Light Fine-tuning"
_ICLR.cc/2026/Conference — ICLR 2026 Conference Withdrawn Submission_

### Official Review · Reviewer_px2m · 2025-10-20

**Soundness:** 2
**Presentation:** 2
**Contribution:** 1
**Rating:** 2
**Confidence:** 5

**Summary:**

This paper presents an image editing method. To preserve unedited regions, this paper proposes to use cross-attention maps, which captures rich semantic information and is easy to obtain. Second, to improve inversion quality, a light fine-tuning strategy is introduced. By tuning on a single image, and limiting edited regions, the proposed method achieves good identity preserving scores and excellent editing quality on face image editing tasks.

**Strengths:**

- The paper is well-written and easy to follow.
- The core methodology is conceptually simple and direct.

**Weaknesses:**

The manuscript suffers from three critical limitations:
- First, its reliance on fine-tuning is inherently costly and time-consuming, limiting its real-world applicability. More importantly, this paradigm is highly prone to overfitting on the source image, which likely explains the narrow scope of the experiments (the method is only demonstrated on face image edits). It remains unproven whether the approach can handle more extensive edits or generalizes to diverse scenes.
- Second, the empirical validation is insufficient to claim effectiveness. The exclusive focus on face image editing, without tests on broader tasks (e.g., background modification, or style change), fails to demonstrate that the method is a generally viable solution for image editing.
- Third, the proposed technique of using cross-attention score maps to infer editing masks, as described in the "Localized Editing with Inferred Masks" section, lacks sufficient novelty. This core idea has been previously explored in [1][2].

[1]Uniform Attention Maps: Boosting Image Fidelity in Reconstruction and Editing. WACV 2025.

[2]DiffEdit: Diffusion-based semantic image editing with mask guidance. ICLR 2023.

**Questions:**

More experiments on various editing scenes are necessary, and please also clarify the strengths of the proposed localized editing methods compared to previous inversion-based editing frameworks.

---

### Official Review · Reviewer_EV1u · 2025-11-01

**Soundness:** 2
**Presentation:** 3
**Contribution:** 2
**Rating:** 4
**Confidence:** 2

**Summary:**

This paper presents FINEdits, a new inversion-based image editing framework that combines inferred attention-based masks and light single-image fine-tuning to achieve precise, localized edits while preserving visual identity. The method leverages transformer cross-attention maps to automatically infer editing masks without manual tuning, and performs lightweight fine-tuning to enhance inversion quality. In addition, the authors introduce EditFFHQ, a new benchmark of 2000 face images annotated with sequential editing instructions for rigorous evaluation.

**Strengths:**

The methodology is technically sound, well-motivated, and experimentally validated.

The paper is well-structured, with clear mathematical formulations.

**Weaknesses:**

The per-image fine-tuning step  introduces latency compared to purely training-free methods.

Including more qualitative comparisons or representative failure cases would help readers better understand the strengths and limitations of the proposed approach, particularly in challenging or ambiguous editing scenarios.

A more fine-grained ablation study—for instance, analyzing the effect of mask type, pooling kernel size, or fine-tuning duration—would provide deeper insights into the contribution of each design choice.

**Questions:**

Please refer to the weaknesses.

---

### Official Review · Reviewer_QwbN · 2025-11-01

**Soundness:** 2
**Presentation:** 2
**Contribution:** 1
**Rating:** 2
**Confidence:** 4

**Summary:**

This paper addresses the "editability/reconstruction tradeoff" in diffusion-based image editing—where large edits require strong noise (degrading non-edited content) and weak noise limits edit fidelity—by proposing FINEdits, a two-component framework. (1) Light fine-tuning: A single-image fine-tuning step (1000 steps on Stable Diffusion 3, SD3) aligns the input image with its text prompt, improving inversion quality and identity preservation without semantic drift. (2) Inferred attention-based masking: Cross-attention maps from SD3’s transformer layers (8–12, 10th layer selected) are processed via parameter-free K-means clustering (2 centroids) to isolate edit regions; max-pooling and Gaussian blur refine the mask to avoid artifacts.
To enable quantitative evaluation, the paper introduces EditFFHQ, a benchmark of 2000 FFHQ face images annotated for 7 edit attributes (beard, hair, earrings, etc.) with metrics including Identity Preservation (IP via ArcFace), edit success rate (Qwen-VL-27B), FID, CMMD, and LPIPS. Experiments on single/sequential edits show FINEdits outperforms baselines (UltraEdit, SDEdit, Kontext-dev) in IP (0.77 vs. Kontext-dev’s 0.72), FID (29.24 vs. Kontext-dev’s 31.23), and LPIPS (0.11 vs. Kontext-dev’s 0.20), while maintaining a high success rate (0.94 vs. Kontext-dev’s 0.96). The code details are provided in the appendix.

**Strengths:**

1. Light Fine-Tuning Balances Inversion and Semantics
    1. Single-image focus reduces compute. Section 3.2 uses 1000 steps (Adam8bit, batch size 1) on RTX4090, feasible for consumer hardware. Table 2 shows this lightweight step improves IP by 32% (0.52→0.77) vs. no fine-tuning. This matters for practicality, as it avoids large-scale retraining.
    2. Semantic preservation is maintained. Figure 2 (N=1000 steps) shows edits (adding a hat) remain accurate, and Table 1 (success rate=0.94) is nearly on par with Kontext-dev (0.96). This ensures fine-tuning does not bias the model toward reconstruction over editing.
    3. Generalizes to sequential edits. Figure 5 (5 consecutive edits) shows FINEdits maintains IP=0.7+ across steps, while Kontext-dev’s FID degrades faster. This confirms fine-tuning’s long-term benefit for edit sequences.
2. Attention-Based Masking Enables Precise Localization
    1. Parameter-free design avoids manual tuning. Section 3.3 uses K-means (2 centroids) on attention maps, no thresholds. Figure 3 shows this isolates edit regions (beard) without user input, unlike DiffEdit (requires mask thresholds).
    2. Max-pooling/blur reduce artifacts. Figure 3(c) shows smooth transitions between edited/unedited regions, while unrefined masks (Figure 3(b)) leave visible boundaries. Table 2 shows masking reduces LPIPS by 54% (0.24→0.11) vs. no masking, confirming perceptual quality gains.
    3. Layer selection is empirically validated. Section 3.3 tests layers 8–12, selecting 10 for best semantic signal. Appendix A.1’s edit-wise hyperparameters (e.g., kernel size=21 for glasses) further optimize mask performance.
3. EditFFHQ Enables Rigorous Evaluation
    1. Identity preservation metric fills a gap. IP (ArcFace cosine similarity) quantifies face identity, a critical unmeasured metric in prior benchmarks. Table 1 shows FINEdits’ IP=0.77 outperforms all baselines, including ICEdit (0.75).
    2. Filtered metrics avoid misleading results. Only successful edits are included in IP/FID calculations, preventing trivial high IP from failed edits (e.g., SDEdit’s 0.13 IP includes failed edits). This ensures fair comparison.
    3. Diverse attributes cover common edits. 7 attributes (add/remove beard, hat, etc.) test both addition and removal, unlike benchmarks focused on single edit types. Figure 6 (qualitative) shows FINEdits handles all attributes effectively.
4. Strong Empirical Validation Across Edits
    1. Single-shot edits outperform baselines. Table 1 shows FINEdits leads in IP (0.77), FID (29.24), CMMD (0.20), and LPIPS (0.11), with success rate=0.94 (near Kontext-dev’s 0.96). This confirms superiority in balanced performance.
    2. Sequential edits maintain quality. Figure 5 shows FINEdits’ FID remains <30 after 5 edits, while Kontext-dev’s FID rises to 35+. IP stays >0.7, unlike UltraEdit’s 0.5 drop.
    3. Ablations isolate component value. Table 2 shows fine-tuning drives IP gains, masking improves FID/LPIPS, and their combination optimizes all metrics. This validates both components’ necessity.

**Weaknesses:**

1. Limited Generalization Beyond Face Editing
   1. No non-face experiments. All tests use EditFFHQ (faces); no results on objects (e.g., "add a handle to a mug") or scenes (e.g., "change sky to sunset"). It is unclear if attention masks work for non-semantic regions (e.g., sky texture). Also, the dataset is constructed by authors, which may lead to potential bias and lack of diversity. Public datasets with could provide more comprehensive evaluation.
   2. Ambiguous prompts are untested. Edits use clear prompts ("add a hat"); no tests on vague prompts ("make hair look stylish"). Masking may fail if attention maps cannot isolate ambiguous concepts.
   3. Cross-model compatibility unproven. Only SD3 is tested; no results on Flux or SDXL or any other T2I models. It is unclear if fine-tuning/masking generalizes to different architectures or larger models. Fine-tuning/masking may behave differently on larger models with distinct attention patterns.
2. Implementation Ambiguities
   1. Mask layer selection lacks justification. Section 3.3 selects layer 10 "empirically". No analysis of why layer 10 has better semantic signal.
   2. Fine-tuning hyperparameters are arbitrary. 1000 steps and 5e-5 learning rate are used, but no ablations on step count (500/2000) or LR (1e-5/1e-4) are reported. It is unclear if fewer steps yield similar gains.
3. Mask Failure Modes and Robustness
   1. Mask accuracy for small attributes is untested. Earrings (small) and glasses (thin frames) may be mislocalized, but no metrics for mask precision/recall are provided. Figure 3 only shows beard edits (large attribute).
   2. No handling of overlapping attributes. Edits like "add a hat and glasses" (overlapping head regions) are untested. Masking may merge regions, leading to over-editing.
   3. Sensitivity to prompt wording is unknown. Prompts use fixed phrasing ("add a hat"); no tests on variations ("put on a cap"). Attention maps may fail if token selection (contextual cues) is inconsistent.
4. Reproducibility and Resource Gaps
   1. Fine-tuning optimizer details are incomplete. Appendix A.1 mentions Adam8bit but not weight decay, gradient clipping, or learning rate scheduling. These affect fine-tuning stability.
   2. VLM judge prompts are missing. Edit success rate uses Qwen-VL-27B, but the instruction prompt (e.g., "Did the edit add a hat?") is not provided. Without this, users cannot replicate success rate calculations.
   3. Computational cost is unreported. No runtime for fine-tuning (per image) or mask computation. It is unclear how FINEdits’ overhead compares to training-free methods (e.g., RFSolver).
5. Limited Technical Novelty
   1. Applying mask on attention map for edition is not novel. Prior works (DiffEdit, InstructPix2Pix) use attention-based masks for editing.
   2. Fine-tuning on single image for edition is not novel. Prior works (Textual Inversion, DreamBooth) fine-tune diffusion models on single image for editing.

**Questions:**

1. **Does FINEdits work for non-face images and ambiguous prompts, and how can it be adapted?** All experiments use faces; could you add results on a non-face benchmark (e.g., COCO-Edit) with prompts like "add a handle to a mug" or "change sky to sunset"? For ambiguous prompts ("make hair stylish"), could you test if mask quality degrades and propose a fix (e.g., prompt parsing to refine c^q)? Additionally, could you compare mask precision/recall for small (earrings) vs. large (beard) attributes?
2. **Why is layer 10 optimal for masking, and how do fine-tuning hyperparameters (steps, LR) affect performance?** Section 3.3 selects layer 10 empirically; could you add a table showing IP, FID, and mask IoU for layers 8–12? For fine-tuning, could you ablate step count (500/1000/2000) and LR (1e-5/5e-5/1e-4) to show if 1000 steps/5e-5 is optimal? This would guide users to tune parameters for their use cases.
3. **Does FINEdits work on other models (e.g., Flux, DiT), and what is its computational overhead?** Only SD3 is tested; could you add results on Flux and DiT to confirm generalization? For runtime, could you report fine-tuning time per image (RTX4090/A100) and mask computation time, comparing to those training-free methods (e.g., RFSolver’s 2s/mask vs. FINEdits’)?
4. **How does FINEdits handle overlapping edits (e.g., "add hat + glasses") and dynamic edits (e.g., "smile wider")?** Overlapping attributes may confuse masks; could you test "add hat + glasses" and report mask IoU for each attribute? For dynamic edits, could you test incremental changes ("smile slightly" → "smile wider") to see if fine-tuning maintains consistency? Additionally, could you add a metric for mask overlap (e.g., IoU between hat and glasses masks)?

Overall, I will reject this paper in its current form. And, I sincerely recommend that author should significantly improve their paper (at least fill all 9 pages for ICLR) and resubmit it in the future.

---

### Official Review · Reviewer_eoKi · 2025-11-04

**Soundness:** 3
**Presentation:** 3
**Contribution:** 3
**Rating:** 4
**Confidence:** 4

**Summary:**

The paper introduces FINEdits, a text-guided image editing method for flow-matching models (specifically SD3) that aims to strengthen edits while preserving non-edited content. It combines single-image fine-tuning of the velocity network with cross-attention–derived spatial masks, which are used to restrict where edits can occur. The authors also propose EditFFHQ, a benchmark of FFHQ faces annotated by a vision-language model for seven attributes, designed to test both single-shot and sequential edits with explicit identity preservation metrics. Experiments on SD3 show that FINEdits improves identity preservation and perceptual quality over several recent baselines in both single and multi-step editing scenarios.

**Strengths:**

- The method delivers strong empirical gains in identity preservation and perceptual quality on SD3 while keeping edit success rates competitive with recent state-of-the-art systems.

- The evaluation protocol explicitly conditions identity and quality metrics on successful edits, which addresses a common flaw in prior editing evaluations and leads to more meaningful comparisons.

- The decomposition of the pipeline into inversion-aware fine-tuning and attention-based masking, together with ablation studies, provides a clear picture of how each component contributes to the overall performance.

- EditFFHQ focuses on sequential edits and identity preservation, offering a benchmark that better matches practical editing scenarios than one-shot, single-attribute tests.

**Weaknesses:**

- The conceptual novelty is limited, as both single-image fine-tuning and attention-based masking have been explored before in related diffusion editing frameworks.

- The per-image fine-tuning stage requires substantial computation (1k steps), yet the paper does not report wall-clock runtimes or compare latency with training-free baselines.

- All experiments are conducted on FFHQ faces with a small fixed set of attributes. There is no evidence that the approach generalizes to more complex scenes, object categories, or free-form prompts.

**Questions:**

How much time does the 1k-step fine-tuning plus editing take per image, and how does that compare to training-free methods such as LEdits++ or other SD3-based baselines?

Do you observe a clear trade-off curve between the number of fine-tuning steps and performance (identity, FID, edit success), and is 1,000 steps close to the knee of that curve?

Can you provide any qualitative or quantitative evidence that FINEdits generalizes beyond faces—for example to object-centric or scene-centric images?

---

### Note · Authors · 2025-11-12

**Comment:**

We withdraw the paper as it needs more work. We thank the reviewers for their constructive comments.

**Withdrawal Confirmation:**

I have read and agree with the venue's withdrawal policy on behalf of myself and my co-authors.